# Structural Requirement of hA5G18 Peptide (DDFVFYVGGYPS) from Laminin α5 Chain for Amyloid-like Fibril Formation and Cell Adhesion

**DOI:** 10.3390/molecules27196610

**Published:** 2022-10-05

**Authors:** Guangrui Zhang, Yuji Yamada, Jun Kumai, Keisuke Hamada, Yamato Kikkawa, Motoyoshi Nomizu

**Affiliations:** Department of Clinical Biochemistry, School of Pharmacy, Tokyo University of Pharmacy and Life Sciences, 1432-1 Horinouchi, Hachioji, Tokyo 192-0392, Japan

**Keywords:** laminin, amyloidogenic peptide, integrin, amyloid-like fibrils, biomaterial

## Abstract

The hA5G18 peptide (DDFVFYVGGYPS) identified from the human laminin α5 chain G domain promotes cell attachment and spreading when directly coated on a plastic plate, but does not show activity when it is conjugated on a chitosan matrix. Here, we focused on the structural requirement of hA5G18 for activity. hA5G18 was stained with Congo red and formed amyloid-like fibrils. A deletion analysis of hA5G18 revealed that FVFYV was a minimum active sequence for the formation of amyloid-like fibrils, but FVFYV did not promote cell attachment. Next, we designed functional fibrils using FVFYV as a template for amyloid-like fibrils. When we conjugated an integrin binding sequence Arg-Gly-Asp (RGD) to the FVFYV peptide with Gly-Gly (GG) as a spacer, FVFYVGGRGD promoted cell attachment in a plate coat assay, but a negative control sequence RGE conjugated peptide, FVFYVGGRGE, also showed activity. However, when the peptides were conjugated to Sepharose beads, the FVFYVGGRGD beads showed cell attachment activity, but the FVFYVGGRGE beads did not. These results suggest that RGD and RGE similarly contribute to cell attachment activity in amyloid-like fibrils, but only RGD contributes the activity on the Sepharose beads. Further, we conjugated a basic amino acid (Arg, Lys, and His) to the FVFYV peptide. Arg or Lys-conjugated FVFYV peptides, FVFYVGGR and FVFYVGGK, showed cell attachment activity when they were coated on a plate, but a His-conjugated FVFYV peptide FVFYVGGH did not show activity. None of the basic amino acid-conjugated peptides showed cell attachment in a Sepharose bead assay. The cell attachment and spreading on FVFYVGGR and FVFYVGGK were inhibited by an anti-integrin β1 antibody. These results suggest that the Arg and Lys residues play critical roles in the interaction with integrins in amyloid-like fibrils. FVFYV is useful to use as a template for amyloid-like fibrils and to develop multi-functional biomaterials.

## 1. Introduction

Laminins, a component of basement membranes, are multifunctional large proteins [1]. Various active sequences from laminin molecules have been identified using synthetic peptides [2]. The active peptides specifically interact with cellular receptors, including integrins and syndecans, a membrane-associated heparan sulfate proteoglycan (HSPG), and are useful to understand the biological function of laminins and to apply in biomaterials and drug delivery systems [3,4]. Previously, we screened biologically active sequences in the human laminin α5 chain G domain using 113 synthetic peptides by a peptide-coated plate and peptide-conjugated chitosan matrix assays [5]. We identified 17 active peptides in both or either the peptide-coated plate and/or peptide-conjugated chitosan matrix assays. Three peptides, hA5G18 (DDFVFYVGGYPS), hA5G26 (LDGTGFARISFD), and hA5G74 (GSLSSHLEFVGI), promoted integrin β1-mediated cell spreading. hA5G74 promoted cell attachment and spreading in both the assays, but hA5G18 and hA5G26 showed activity only in the peptide-coated plate assay. These results suggested that hA5G18 and hA5G26 aggregate on the plastic plates and promote cell adhesion, but the mechanism of the aggregation has not been identified.

The amyloid fibril formation of degraded proteins is often related to diseases, such as Alzheimer’s disease, type II diabetes, Parkinson’s disease, prion diseases, and systemic polyneuropathies [6,7,8]. The identification of amyloidogenic peptides leads to a better understanding of the mechanism of disease [6,7,8,9]. Detailed studies have also been conducted on the structural characteristics of amyloid β (Aβ) oligomers and fibrils in the neural tissue of Alzheimer’s disease, and it has been reported that differences in morphology, physicochemical properties, and cytotoxicity occur due to their structural diversity [10]. It is also known that peptides with amyloid fibrils are often arranged to form β-strands and specifically bind dyes, such as thioflavin T and Congo red [11,12]. Previously, we identified diverse amyloidogenic peptides from laminin sequences [13,14]. All of the peptides were stained with Congo red and most of peptides showed amyloid-like fibril formation in an electron microscopic analysis. We also showed that many of the amyloidogenic peptides contain basic amino acids and cellular effects, including the promotion of cell attachment and neurite outgrowth [13]. The amyloidogenic peptides are useful to understand the effect of the degraded proteins in vivo. Further, amyloidogenic peptides have been modified with biologically active peptides and been applied in multifunctional biomaterials [15]. Amyloidogenic peptides are useful to understand the function of degraded proteins and to apply in biomaterials. 

In this paper, we focused on two peptides, hA5G18 and hA5G26, and identified an essential core sequence of hA5G18 for amyloid-like fibril formation. We also design functional fibrils using the core sequence of hA5G18.

## 2. Result

### 2.1. Amyloidogenic Peptides from the Human Laminin α5 Chain G Domain

hA5G18 and hA5G26, derived from the human laminin α5 chain G domain, promoted integrin-mediated cell attachment and spreading only in a peptide-coated plate assay [5]. We focused on two peptides, hA5G18 and hA5G26, and evaluated their amyloidogenicity. B133, an amyloidogenic peptide that promotes integrin-mediated cell adhesion [14], was used as a positive control. We evaluated the effect of the peptides on the Congo red absorption spectrum (Figure 1A). Congo red binds to amyloid fibrils and promotes an absorption peak shift from 490 to 540 nm [16]. The Congo red absorption peak at 490 nm was significantly shifted at 540 nm by hA5G18, similar to that by B133, but was not influenced by hA5G26. We also examined the fibril formation of the peptides using a TEM (Figure 1B). hA5G18 exhibited typical amyloid-like fibrils similar to those of B133, but hA5G26 did not form fibrils. These results indicate that hA5G18 forms amyloid-like fibrils.

### 2.2. Amyloid-like Fibril Formation and Cell Attachment Activity of Truncated hA5G18 Peptides 

We synthesized truncated peptides of hA5G18 to identify active core sequences for amyloid-like fibril formation and cell attachment (Table 1, Figure 2). AG73 (RKRLQVQLSIRT) from the laminin α1 chain G domain was used as a positive control for the cell attachment assay [17]. First, the N-terminally truncated hA5G18 peptides were examined by the Congo red staining, a TEM analysis, and a cell attachment assay. When the Congo red solution was incubated with the truncated peptides, the absorption peak was shifted to a long wavelength by hA5G18A and hA5G18B, but the hA5G18C peptide, N-terminal with three amino acids deleted, did not influence the absorption spectrum (Figure 2A). In addition, hA5G18A and hA5G18B showed fibrils in a TEM, but hA5G18C did not (Figure 2B). Further, hA5G18A and hA5G18B promoted cell attachment in a dose-dependent manner similar to that of hA5G18 and AG73, but hA5G18C did not (Figure 2C). These results suggest that hA5G18B has both amyloid-like fibril formation and cell attachment activity similar to those of hA5G18.

Next, C-terminally truncated hA5G18B peptides were examined by the Congo red staining, a TEM analysis, and a cell attachment assay (Table 1). The hA5G18BTC5 peptide, C-terminal five amino acids deleted, still showed Congo red staining, but further shortened peptide hA5G18BTC6 eliminated the activity (Figure 2A). In addition, hA5G18BTC5 showed fibrils in a TEM analysis, but hA5G18BTC6 did not (Figure 2B). These results suggest that hA5G18BTC5 (FVFYV) is a minimum active sequence for amyloid-like fibril formation. In contrast, none of the C-terminally truncated hA5G18B peptides showed cell attachment activity (Figure 2C), suggesting that hA5G18B (FVFYVGGYPS) is a minimum active sequence for cell attachment.

### 2.3. Cell Attachment Activity of hA5G18B Conjugated Sepharose Bead

Next, we prepared an hA5G18B-conjugated Sepharose bead and tested the cell attachment activity to evaluate whether amyloid-like fibril formation was a requirement for the activity. An AG73 bead was used as a control [3]. The AG73 bead showed cell attachment, but the hA5G18B bead did not (Figure 2D). These results suggest that the cell attachment activity of hA5G18B requires amyloid-like fibril formation.

### 2.4. Congo Red Analysis and Cell Attachment Activity of Alanine-Substituted hA5G18B Peptides 

We synthesized alanine-substituted hA5G18B peptides to evaluate critical amino acids for cell attachment and amyloid-like fibril formation (Table 2, Figure 3). When the Val5, Gly6, Pro9, and Ser10 residues of hA5G18B were substituted with Ala, the peptides influenced the absorption spectrum of Congo red as well as that of hA5G18B (Figure 3A). Additionally, the Congo red absorption peak was weakly shifted by hA5G18BA7(G) and hA5G18BA8(Y). In contrast, hA5G18BA1(F), hA5G18BA2(V), hA5G18BA3(F), and hA5G18BA4(Y) did not influence the absorption spectrum of Congo red. These results suggest that the Phe1, Val2, Phe3, and Tyr4 residues are critical and the Gly7 and Tyr8 residues partially contribute to the amyloid-like fibril formation of hA5G18B.

Next, we evaluated the cell attachment activities of alanine-substituted hA5G18B peptides (Table 2, Figure 3B). The cell attachment activity was significantly decreased when the Phe1, Val2, Phe3, and Tyr4 residues were substituted with Ala (Figure 3B). The other alanine substitutions did not influence the cell attachment activity. These results suggest that the Phe1, Val2, Phe3, and Tyr4 residues are critical for cell attachment activity.

The cell morphology of alanine-substituted hA5G18B peptides was examined (Figure 4). AG73 was used as a control. hA5G18B promoted cell spreading and AG73 did not. A5G18BA5(V), hA5G18BA6(G), hA5G18BA7(G), hA5G18BA8(Y), and hA5G18BA10(S) promoted cell spreading. In contrast, hA5G18BA9(P) showed cell attachment but did not promote cell spreading. These results suggest that the Pro9 residue is involved in the cell spreading activity of A5G18B.

### 2.5. Conjugation of an Arg-Gly-Asp (RGD) Sequence and Basic Amino Acids to FVFYV

FVFYV formed amyloid-like fibrils and did not promote cell attachment (Figure 2). Next, FVFYV was used as an amyloid-like fibril template and modified with a cell adhesive peptide. We conjugated an integrin binding sequence Arg-Gly-Asp (RGD) [18] to FVFYV with Gly-Gly as a spacer. We also conjugated Arg-Gly-Glu (RGE), a negative control sequence of RGD, to FVFYV similarly as a control. FVFYVGGRGD and FVFYVGGRGE were stained with the Congo red and were suggested to form amyloid-like fibrils (Figure 5A) (Table 3). FVFYVGGRGD showed cell attachment activity in a dose-dependent manner (Figure 5B). However, FVFYVGGRGE also promoted cell attachment similarly. These results suggest that the cell attachment activity of FVFYVGGRGD in the peptide coated-plate assay is not due to the integrin binding sequence RGD, and the Arg residue in amyloid-like fibrils may cause the activity.

Various amyloidogenic peptides have been identified and many of the peptides promote cell attachment [13]. Additionally, most of the cell adhesive amyloidogenic peptides contain basic amino acids [13]. To evaluate effect of basic amino acids on the cell attachment activity of amyloid-like fibrils, we conjugated Arg, Lys, and His to the C-terminus of FVFYV with Gly-Gly as a spacer (Table 3). When the Congo red solution was incubated with the peptides, the absorption peak was shifted by all the peptides. These results suggest that the basic amino acid-conjugated FVFYV peptides form amyloid-like fibrils (Figure 5A). We also evaluated the cell attachment activity of the peptides (Figure 5B). FVFYVGGR and FVFYVGGK promoted cell attachment in a dose-dependent manner. In contrast, FVFYVGGH did not promote cell attachment, similarly to FVFYV. These results suggest that Arg and Lys residues contribute to the cell attachment activity in amyloid-like fibrils. 

Next, we examined the cell attachment activity of FVFYVGGRGD, FVFYVGGRGE, FVFYVGGR, FVFYVGGK, FVFYVGGK, and RGGFVFYV in the Sepharose bead assay (Figure 5C). AG73 was used as a control. The AG73 bead showed cell attachment activity as shown previously [3]. The FVFYVGGRGD bead promoted cell attachment, but the other peptide-beads did not show the activity.

### 2.6. Effect of EDTA and Heparin on Cell Attachment to Peptide-Coated Plates 

We examined the effect of EDTA and heparin on the cell attachment to FVFYVGGRGD, FVFYVGGRGE, FVFYVGGR, and FVFYVGGK to determine the cellular receptors in a peptide-coated plate assay (Figure 6). Additionally, we also examined poly arginine (poly-R). EF1 was used as the EDTA inhibition control to interact with integrin α2β1 and promote divalent cation-dependent cell adhesion [19]. AG73 was used as a control to interact with syndecans and promote heparin-dependent cell attachment [20]. Cell attachment to EF1 was inhibited by EDTA but not by heparin, and that to AG73 was inhibited by heparin but not by EDTA, as shown previously [19]. Cell attachment to FVFYVGGRGD, FVFYVGGRGE, FVFYVGGR, FVFYVGGK, and poly-R was inhibited by EDTA and heparin. These results suggest that the cell attachment of FVFYVGGR, FVFYVGGK, FVFYVGGRGD, FVFYVGGRGE, and poly-R is mediated by integrins and HSPGs.

### 2.7. Effect of Anti-Integrin Antibodies on Cell Attachment and Spreading to Peptide-Coated Plates

We examined effect of anti-integrin antibodies on the cell attachment to FVFYVGGR, FVFYVGGK, FVFYVGGRGD, FVFYVGGRGE, and poly-R (Figure 7). EF-1 was used as a control. We used antibodies against integrin αvβ3, α2β1, α3/α6, and β1 [21,22]. The cell attachment to EF-1 was inhibited by anti-integrin α2β1 and β1 antibodies, as shown previously. Additionally, the cell attachment to EF-1 was weakly inhibited by the anti-integrin α3/α6 antibody [19]. The cell attachment to FVFYVGGRGD, FVFYVGGRGE, FVFYVGGR, FVFYVGGK, and poly-R was significantly inhibited by the anti-integrin β1 antibody, while the other anti-integrin antibodies did not influence the cell attachment (Figure 7). These results suggest that integrin β1 is involved in the cell attachment to FVFYVGGRGD, FVFYVGGRGE, FVFYVGGR, FVFYVGGK, and poly-R.

Next, we examined effect of an integrin antibody on the cell morphology. The cell spreading on EF-1, FVFYVGGRGD, FVFYVGGRGE, FVFYVGGR, FVFYVGGK, and poly-R was significantly inhibited by the anti-integrin β1 antibody (Figure 8). Additionally, the cell spreading on the all peptides, including FVFYVGGRGD, was not inhibited by the anti-αvβ3 and anti-integrin α3/α6 antibodies (data not shown). These results suggest that FVFYVGGRGD, FVFYVGGRGE, FVFYVGGR, FVFYVGGK, and poly-R promote integrin β1-mediated cell spreading.

## 3. Discussion 

Amyloid fibrils are in many cases observed to interact with cells and are often related to diseases. Here, we focused on the amyloid-like fibril formation of laminin peptide fragments. We previously identified 12 biologically active sequences in the human laminin α5 G domain using 115 synthetic peptides, as well as peptide-coated plates and a peptide-conjugated chitosan assay [5]. Two peptides, hA5G18 and hA5G26, promoted cell attachment and spreading only in the plate assay [5]. We described how hA5G18 forms amyloid-like fibrils by a Congo red staining assay and an electron microscopic analysis. hA5G18 was active in the plate assay but inactive in the bead assay. These results suggest that amyloid-like fibril formation is required for the cell attachment activity of hA5G18. The deletion analysis showed that FVFYV is a minimal essential sequence for amyloid-like fibril formation but is not active in cell attachment. FVFYV has a potential to be used as a core sequence for amyloid-like fibrils. 

We designed a functional amyloid-like fibril using an integrin αvβ3 binding sequence RGD and FVFYV with GG as a spacer. The RGD conjugated the FVFYV peptide FVFYVGGRGD and the negative control FVFYVGGRGE formed amyloid-like fibrils and similarly promoted cell attachment. Further, the cell adhesion and spreading activities of the peptides were not influenced by the anti-integrin αvβ3 antibody but were inhibited by the anti-integrin β1 antibody. These results suggest that RGD and RGE contribute similarly in the fibrils and the Arg residue may be involved in the activity. Previously we described that basic amino acids in amyloidogenic peptides are essential for cell attachment activity through HSPGs, such as syndecans [13,14]. We conjugated basic amino acids (Arg, Lys, His) to the FVFYV peptide with GG as a spacer and examined their activity (Table 3). The three peptides were stained with Congo red. FVFYVGGR and FVFYVGGK promoted cell attachment activity, but FVFYVGGH did not. These results suggest that His residue is weakly basic and does not contribute the activity. Additionally, FVFYVGGR and FVFYVGGK promote integrin-mediated cell attachment. These results suggest that Arg and Lys residues promote cellular effects when they are incorporated in amyloid-like peptide fibrils. However, polymerized basic amino acids, such as poly-R, promoted integrin β1-mediated cell adhesion similar to that of FVFYVGGR and FVFYVGGK. The cell attachment of poly-R was completely inhibited by heparin and weakly inhibited by the EDTA and anti-integrin β1 antibody. However, the cell spreading of poly-R was completely inhibited by the anti-integrin β1 antibody. Since poly-R is highly basic and interacts strongly with HSPG, it may be difficult to find the effect of EDTA and the anti-integrin β1 antibody. Polymerized basic amino acids in polymer or fibrils have a potential to interact with heparin and integrin β1 [18]. Various amyloidogenic peptides have previously been identified [13]. Many amyloidogenic peptides contain Arg and Lys residues and have cell attachment activity. The Arg and Lys residues may contribute cellular effects as a basic cluster in the fibrils. 

hA5G18B (FVFYVGGYPS) is a minimum sequence for amyloid-like fibril formation and cell attachment. The Ala-substitution analysis of hA5G18B suggests that the Pro residue is critical for the cell spreading activity of hA5G18B. However, the mechanism of Pro is not clear at this time. These results suggest that sequences other than basic amino acids may be active in amyloid fibrils.

Self-assembling amyloidogenic peptides are easily modified with biologically active ligands, including peptides, and functional amyloid-like fibrils have been used as a biomaterial [15,23]. The FVFYV peptide assembles itself and forms amyloid-like fibrils without cellular effects and is easily modified with functional sequences. FVFYV is useful to apply as a core sequence for designing functional amyloid-like fibrils and has a potential for use as a unique platform for a cell scaffold material.

## 4. Materials and Methods 

### 4.1. Synthetic Peptides 

All peptides were synthesized by the 9-fluorenylmethoxycarbonyl (Fmoc)-based solid-phase method with a C-terminal amide. The Rink amide resin ((4,2,4-dimethoxyphenyl-Fmoc-aminomethyl)-phenoxy resin) was weighed on a 50 µmol scale. The weighed resin was placed in PD-10 column (Cytiva, Tokyo, Japan) and deprotection of the Fmoc group to the resin was carried out by adding 20% piperidine containing N,N-dimethylformamide (DMF) and shaking for 15 min at room temperature. The resin was washed three times with DMF and reacted with an Fmoc-protected amino acid in DMF using N,N′-Diisopropylcarbodiimide (DIC)/Hydroxybenzotriazole (HOBt) method for 1 h. Afterwards, Fmoc deprotection and coupling of each amino acid were repeated, and finally the resin was washed with diethyl ether and dried. To obtain crude peptide, trifluoroacetic acid (TFA): *m*-cresol: ethane-1,2-dithiol: thioanisole: H_2_O (80:5:5:5:5, *v*/*v*/*v*/*v*/*v*) solution was added and incubated for 3 h at room temperature. The crude peptides were purified by reversed phase HPLC using Mightysil RP-18 GP 250-20 column (Kanto Chemical Co., Inc., Tokyo, Japan) with a binary solvent system (0.1% TFA in acetonitrile and 0.1% TFA in water). The purified peptides solutions were lyophilized and obtained purified peptide as white feathery powders. Purity and mass of the peptides were confirmed by an analytical HPLC and an electrospray ionization mass spectrometer at the Central Analysis Center, Tokyo University of Pharmacy and Life Sciences.

### 4.2. Antibodies

Rat monoclonal antibody against human integrin α6 (P5G10) was purchased from AMAC (Westbrook, ME, USA). Mouse monoclonal antibodies against human integrin αvβ3 (VNR-1), α3 (P1B5), β1 (AIIB2), and α2β1 (VLA-2) were purchased from Millipore Co., Ltd. (Billerica, MA, USA). Mouse monoclonal antibody against human IgG heavy chain (MR36G) was purchased from Sigma-Aldrich (St. Louis, MO, USA).

### 4.3. Congo Red Binding Analysis

To make a 100 μM Congo red stock solution, Congo red was dissolved in 10% ethanol containing phosphate-buffered saline (PBS) and filtered three times using a 0.45-micron nylon membrane (Iwaki Co., Ltd., Tokyo, Japan). Each of the peptide solutions (0.1 mM, 100 µL) in Mill-Q water and the Congo red stock solution were mixed with 800 μL of 1.25× PBS and incubated in disposable cuvettes for 24 h at room temperature in the dark. Absorption spectra were measured from 300 to 700 nm using a UV-1700 UV/Vis spectrophotometer (Shimadzu Co., Ltd., Kyoto, Japan).

### 4.4. Transmission Electron Microscopy (TEM)

Peptide solution (1 mM) was diluted 1:0 to 1:4 in Milli-Q water and applied onto a grid mesh with carbon-coated Formvar film (Okenshoji Co., Ltd., Tokyo, Japan). The specimen was negatively stained with a 2% aqueous solution of phosphotungstic acid and observed using JEM-1011 (JEOL Ltd., Tokyo, Japan) electron microscope at an acceleration voltage of 80 kV.

### 4.5. Cells and Culture

Human neonatal dermal fibroblasts (HDFs) (AGC Techno Glass Co., Ltd., Chiba, Japan) were maintained in low glucose containing Dulbecco’s modified Eagle medium (DMEM; Invitrogen, Carlsbad, CA, USA) with 10% fetal bovine serum (FBS, Invitrogen), 100 U/mL penicillin, and 100 μg/mL streptomycin (Invitrogen) under 37 °C in a humidified, 5% CO_2_ atmosphere.

### 4.6. Cell Attachment Assay Using Peptide-Coated Plates

96-well plates (Nunc, Inc., Naperville, IL, USA) were coated with various amounts of peptides in water and dried over at room temperature. After washing three times with PBS, the peptide-coated wells were blocked with 1% bovine serum albumin (BSA; Sigma, St. Louis, MO, USA) in DMEM (150 μL) for 1 h, then washed twice with DMEM containing 0.1% BSA. Cells were detached by 0.05% trypsin/ethylenediaminetetraacetic acid (EDTA) and resuspended in DMEM containing 0.1% BSA and seeded to each well (20,000 cells/100 μL) and incubated for 1 h at 37 °C. The attached cells were stained with 0.2% crystal violet aqueous solution containing 20% methanol for 10 min at room temperature. After washing several times with water, the attached cells were photographed using a BZ-X810 microscope (Keyence, Osaka, Japan). Images were analyzed using BZ-analyzer software (Keyence). The attached cells in three randomly selected fields were counted. Each experiment repeated at least three times.

### 4.7. Inhibition Assay

For inhibition of cell attachment to peptide-coated plates, 96-well plates were coated with peptides as described above. Poly-L-arginine hydrochloride (molecular weight > 70,000) was dissolved in Mill-Q water (final concentration of 2 mM), coated, and dried over at room temperature. Cells were preincubated for 15 min at 37 °C in the presence of either 10 mg/mL heparin, 5 mM EDTA, 10 μg/mL of the anti-integrin αvβ3, α3/α6, β1 antibodies or 30 μg/mL of the anti-integrin α2β1 antibody. HDFs (2.0 × 10^4^ cells/100 μL) were incubated at 37 °C in 5% CO_2_ on the peptide-coated plates for 30 min. After washing several times with PBS (-), attached cells were stained with 0.2% crystal violet in 20% methanol for 15 min, and imaged by the BZ-X810 microscope. The figures of attached cells in nine central fields (0.77 mm^2^ each) were counted and their average number was calculated using BZ-X800 Analyzer software. The area of the attached cells was measured using BZ-X800 Analyzer software.

### 4.8. Cell Attachment Assay Using Peptide-Conjugated Sepharose Beads

The synthetic peptides were coupled to cyanogen bromide (CNBr)-activated Sepharose 4B (GE Healthcare, London, UK) as described previously [24]. The peptides (200 μg) were incubated with the CNBr-activated Sepharose (30 mg) in 1 mL of 0.1 M NaHCO_3_ solutions containing 0.5 M NaCl (pH 8.3) for 1 h at room temperature. Then, unreacted CNBr residues were quenched with 1 M ethanolamine (pH 8.0) and washed with 0.1 M acetate buffer (pH 4.0) containing 0.5 M NaCl and 0.5 M NaCl with 0.1 M Tris-HCl buffer (pH 8.0). HDF cells were resuspended in 0.1% BSA containing DMEM (1 × 10^5^ cells/100 μL) and incubated with 3 mg/50 μL peptide-beads solution for 1 h at 37 °C. The attached cells were stained with 0.2% crystal violet aqueous solution in 20% methanol. After removal of unattached cells, the attached cells were observed under a BZ-X810. Each assay was repeated at least twice.

### 4.9. Statistics Analysis

Results were expressed as ± standard deviation (S.D.). Comparison of mean values was performed using one-way analysis of variance and a homoscedastic *t* test. *p* < 0.05 indicated statistical significance. 

## Figures and Tables

**Figure 1 molecules-27-06610-f001:**
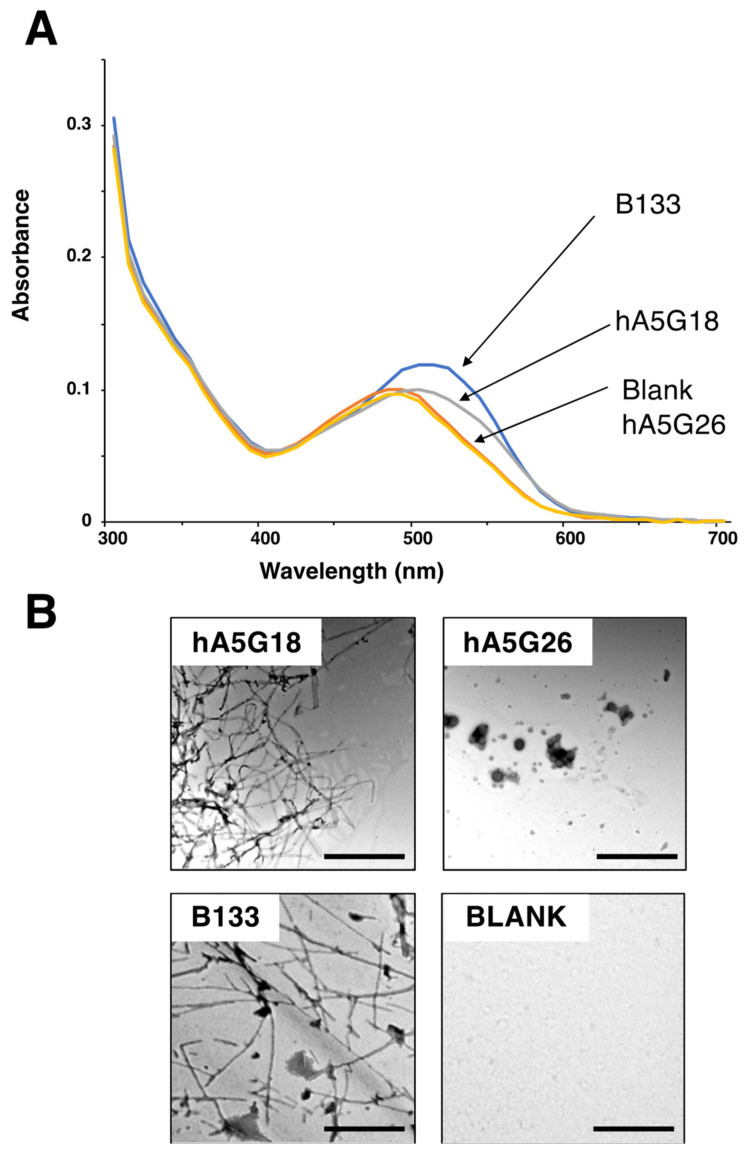
Congo red staining and electron micrographic analysis of peptides. (**A**) Absorption spectra of peptides with Congo Red were recorded from 300 to 700 nm. (**B**) Electron micrograph of amyloid-like fibrils formed from peptides. Peptide solution (1 mM) was diluted 1:0 to 1:4 with water and smear solution on a grid mesh with carbon-coated Formvar film. Then the specimen was negatively stained with a 2% aqueous solution of uranyl acetate and observed using an electron microscope. Bar = 500 nm.

**Figure 2 molecules-27-06610-f002:**
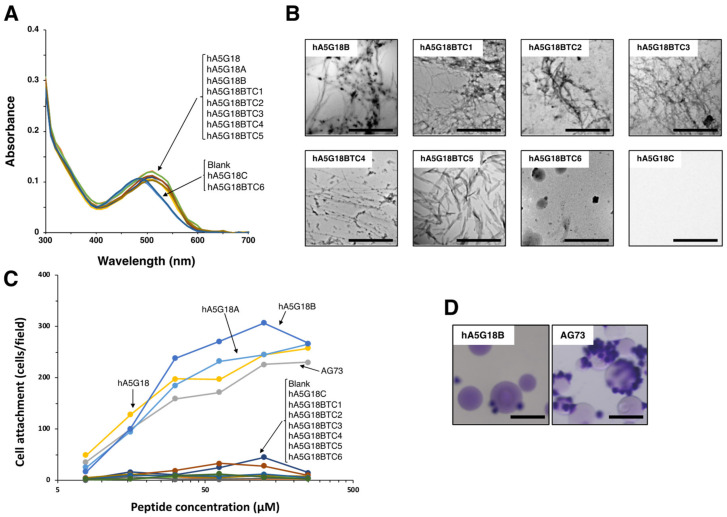
Amyloid-like fibril formation and cell attachment activity of the truncated hA5G18 peptides. (**A**) Peptides were stained with Congo red and absorption spectra were recorded from 300 to 700 nm. (**B**) Electron micrograph of peptides. Bars = 500 nm. (**C**) Cell attachment of the truncated hA5G18 peptides in a peptide-coated plate assay. Peptide-coated plates were prepared as described in the Materials and Method section and HDFs (2 × 10^4^ cells/well) were added and incubated for 1 h. The data are expressed as the means of triplicate results. Triplicate experiments gave similar results. (**D**) Cell attachment of peptide-Sepharose beads. HDFs were allowed to attach to peptide-Sepharose beads for 1 h and then stained with 0.2% crystal violet in 20% methanol. Bar = 100 nm.

**Figure 3 molecules-27-06610-f003:**
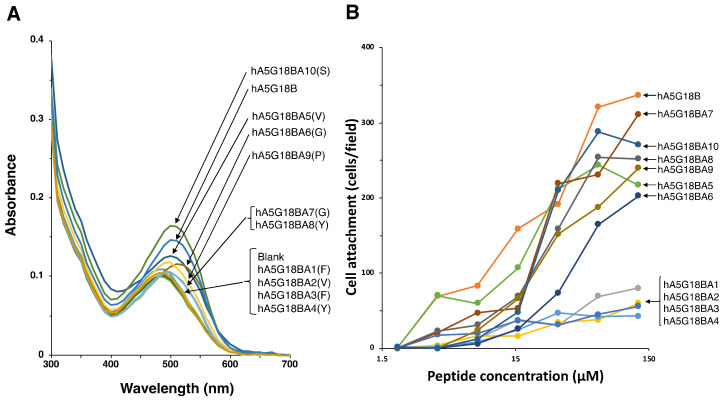
Amyloidogenicity and biological activity of the Ala-substituted hA5G18B peptides. (**A**) Absorption spectra of peptides stained with Congo Red. Peptide solution (100 μL, 1 mM in H_2_O) and Congo red solution (100 μL, 100 μM in PBS) were mixed with 800 μL of PBS and incubated for 24 h at room temperature. Absorption spectra were recorded from 300 to 700 nm. (**B**) Cell attachment activity of the Ala-substituted hA5G18B peptides. Peptide-coated plates were prepared as described in the Materials and Method section and HDFs (2 × 10^4^ cells/well) were added and incubated for 1 h. After being stained with 0.2% crystal violet in 20% methanol, the number of the attached cells was counted. The data are expressed as the means of triplicate results.

**Figure 4 molecules-27-06610-f004:**
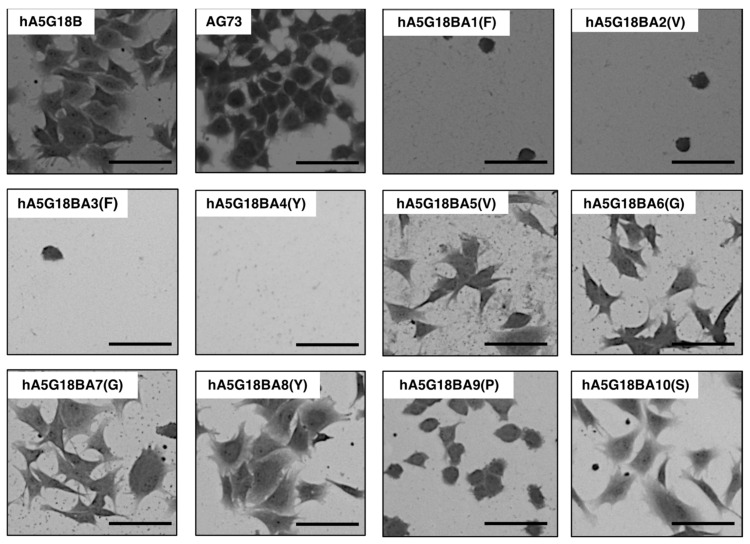
Morphological appearance of HDFs on peptide-coated plates. Peptides (2 nmol/well of hA5G18B and hA5G18BA1-10, 0.4 nmol/well of AG73) were coated on plates and HDFs (2 × 10^4^ cells/well) were added. After a 1 h incubation, cells were stained with 0.2% crystal violet in 20% methanol. Bar = 100 nm.

**Figure 5 molecules-27-06610-f005:**
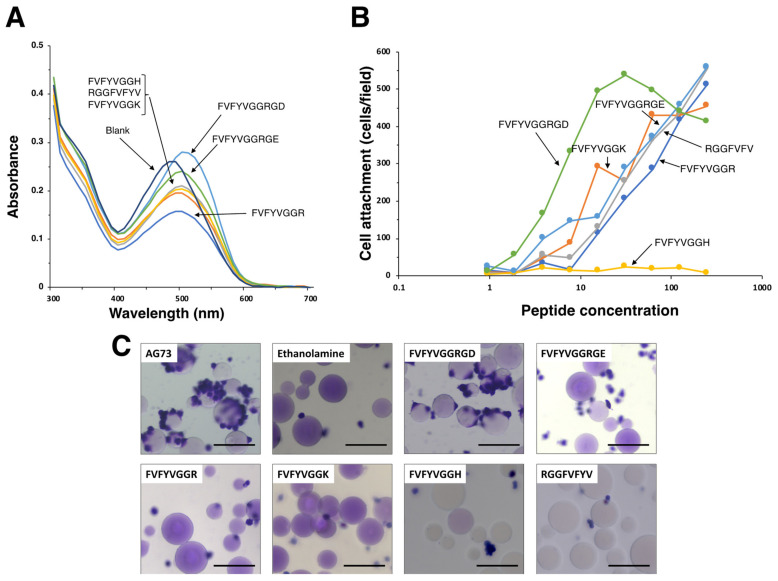
Amyloidogenicity and biological function of modified FVFYV peptides. (**A**) Absorption spectra of peptides stained with Congo Red. Peptide solution (100 μL, 1 mM in H_2_O) and Congo red solution (100 μL, 100 μM in PBS) were mixed with 800 μL of PBS and incubated for 24 h at room temperature. Absorption spectra were recorded from 300 to 700 nm. (**B**) HDFs attachment to peptide-coated plates. Peptide-coated plates were prepared as described in the Materials and Method section. HDFs (2 × 10^4^ cells/well) were added to the wells and incubated for 1 h. After being stained with with 0.2% crystal violet in 20% methanol, the number of the attached cells was counted. The data are expressed as the means of triplicate results. Triplicate experiments gave similar results. (**C**) Cell attachment to the peptide-Sepharose beads. HDFs were allowed to attach to the peptide-Sepharose beads for 1 h and then were stained with 0.2% crystal violet in 20% methanol. Bar = 100 nm.

**Figure 6 molecules-27-06610-f006:**
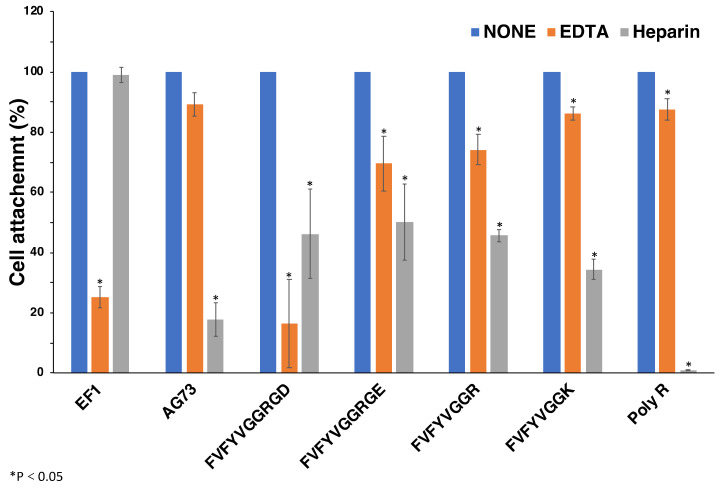
Effect of EDTA and heparin on cell attachment to the peptides. 96-well plate was coated with peptides (2 nmol/well of EF1, 0.4 nmol/well of AG73, 5 nmol/well of FVFYVGGRGD and FVFYVGGRGE, 10 nmol/well of FVFYVGGR and FVFYVGGK, 1 ng/well of poly R). HDFs were mixed with either 5 mM EDTA or 10 μg/mL heparin and then added to the plates. After a 1 h incubation, cells were stained with crystal violet and the number of the attached cells was counted. Each value represents the mean of three separate determinations ± S.D. Triplicate experiments gave similar results. * *p <* 0.05.

**Figure 7 molecules-27-06610-f007:**
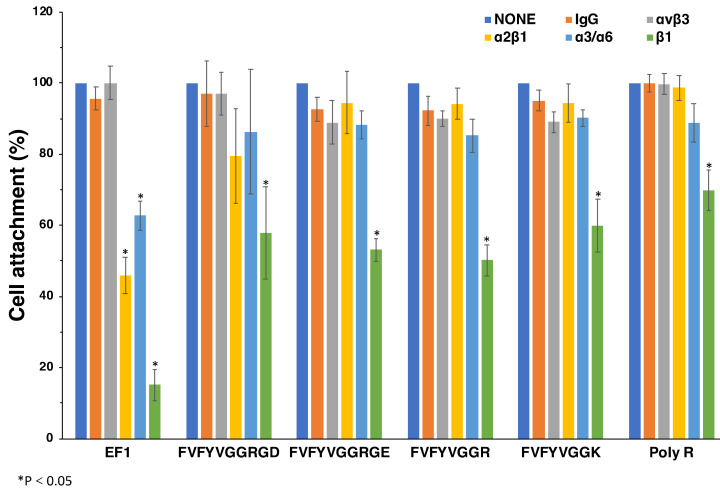
Effect of anti-integrin antibodies on cell attachment to peptides. HDFs were preincubated with 10 μg/mL of the integrin antibodies at room temperature for 15 min and added to the peptide-coated plates (2 nmol/well of EF1, 5 nmol/well of FVFYVGGRGD and FVFYVGGRGE, 10 nmol/well of FVFYVGGR and FVFYVGGK, 1 ng/well of poly R). After a 1 h incubation, cells were stained with crystal violet and the number of the attached cells was counted. Each value represents the mean of three separate determinations ± S.D. Triplicate experiments gave similar results. * *p <* 0.05.

**Figure 8 molecules-27-06610-f008:**
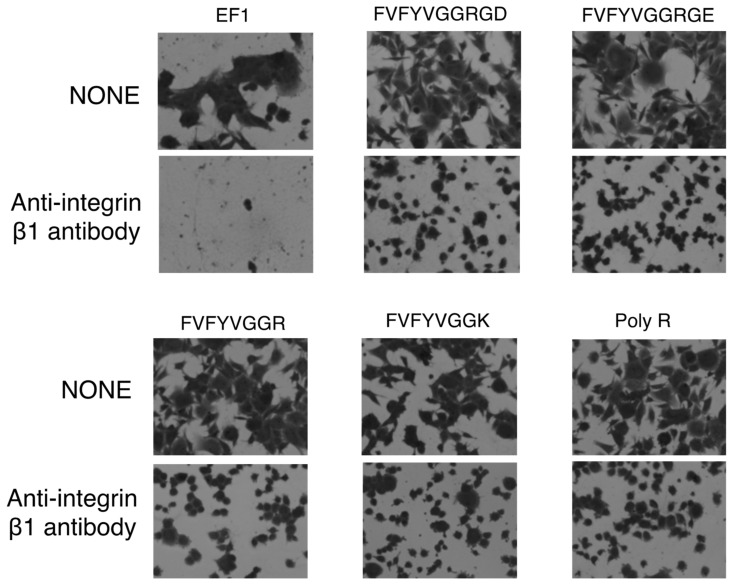
Effect of anti-integrin β1 antibody on cell adhesion to peptides. HDFs were preincubated with 10 μg/mL of integrin β1 antibody at room temperature for 15 min and added to the peptide-coated plates (2 nmol/well of EF1, 5 nmol/well of FVFYVGGRGD and FVFYVGGRGE, 10 nmol/well of FVFYVGGR and FVFYVGGK, 1 ng/well of poly R). After incubation for 1 h, cells were stained with 0.2% crystal violet in 20% methanol.

**Table 1 molecules-27-06610-t001:** Biological activities of hA5G18 and truncated hA5G18 peptides.

Peptide	Sequence	Congo Red Staining ^a^	Cell Attachment ^b^	Cell Spreading ^c^
hA5G18	DDFVFYVGGYPS	+	+	+
hA5G18A	DFVFYVGGYPS	+	+	+
hA5G18B	FVFYVGGYPS	+	+	+
hA5G18C	VFYVGGYPS	-	-	-
hA5G18BTC1	FVFYVGGYP	+	-	-
hA5G18BTC2	FVFYVGGY	+	-	-
hA5G18BTC3	FVFYVGG	+	-	-
hA5G18BTC4	FVFYVG	+	-	-
hA5G18BTC5	FVFYV	+	-	-
hA5G18BTC6	FVFY	-	-	-

^a^ Peptides were incubated with a Congo red solution, and the absorption spectra measuring from 300 to 700 nm was evaluated on the following subjective scale: + showed absorption peak shift; and - no shift in the absorption peak. ^b^ Cell attachment activity was scored on the following subjective scale: + showed cell attachment activity; and - no activity. ^c^ Cell spreading activity was scored on the following subjective scale: + showed cell spreading activity; and - no activity.

**Table 2 molecules-27-06610-t002:** Biological activities of hA5G18B and its alanine-substituted derivatives.

Peptide	Sequence ^a^	Congo Red Staining ^b^	Amyloid-like Fibril Formation ^c^	Cell Attachment ^d^	Cell Spreading ^e^
hA5G18B	FVFYVGGYPS	+	+	+	+
hA5G18BA1(F)	**A**VFYVGGYPS	-	-	-	-
hA5G18BA2(V)	F**A**FYVGGYPS	-	-	-	-
hA5G18BA3(F)	FV**A**YVGGYPS	-	-	-	-
hA5G18BA4(Y)	FVF**A**VGGYPS	-	-	-	-
hA5G18BA5(V)	FVFY**A**GGYPS	+	+	+	+
hA5G18BA6(G)	FVFYV**A**GYPS	+	+	+	+
hA5G18BA7(G)	FVFYVG**A**YPS	+	+	+	+
hA5G18BA8(Y)	FVFYVGG**A**PS	+	+	+	+
hA5G18BA9(P)	FVFYVGGY**A**S	+	+	+	-
hA5G18BA10(S)	FVFYVGGYP**A**	+	+	+	+

^a^ Substituted alanine is shown in bold. ^b^ Peptides were incubated with a Congo red solution, and the absorption spectra measuring from 300 to 700 nm was evaluated on the following subjective scale: + showed absorption peak shift; and - no shift in the absorption peak. ^c^ Peptides were examined by TEM and evaluated on the following subjective scale: + showed amyloid-like fibrils; and - no fibrils. ^d^ Cell attachment activities were scored on the following subjective scale: + showed cell attachment activity; and - no activity. ^e^ Cell spreading activity was scored on the following subjective scale: + showed cell spreading activity; and - no activity.

**Table 3 molecules-27-06610-t003:** Biological activities of modified FVFYV peptides.

Sequence	Congo Red Staining ^a^	Cell Attachment ^b^	Cell Spreading ^c^
FVFYVGGRGD	+	+	+
FVFYVGGRGE	+	+	+
FVFYVGGR	+	+	+
FVFYVGGK	+	+	+
FVFYVGGH	+	-	-
RGGFVFYV	+	+	+

^a^ Peptides were incubated with a Congo red solution, and the absorption spectra measuring from 300 to 700 nm was evaluated on the following subjective scale: + showed absorption peak shift. ^b^ Cell attachment activities were scored on the following subjective scale: + showed cell attachment activity; and - no activity. ^c^ Cell spreading activity was scored on the following subjective scale: + showed cell spreading activity; and - no activity.

## Data Availability

Not applicable.

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
