# Peer review of "Structural Requirement of hA5G18 Peptide (DDFVFYVGGYPS) from Laminin α5 Chain for Amyloid-like Fibril Formation and Cell Adhesion"

_molecules, 2022, doi:10.3390/molecules27196610_

Round 1

Reviewer 1 Report

I carefully read the paper by Nomizhu and colleagues since it contains intriguing insights. Here, the authors first concentrated on the two peptides hA5G18 and hA5G26 and found that hA5G18's core sequence is critical for the formation of amyloid-like fibrils. When directly coated on a plastic plate, the hA5G18 peptide (DDFVFYVGGYPS), which was isolated from the human laminin 5 chain G 9 domain, increases cell adhesion and spreading, but does not exhibit the same activity when conjugated on a chitosan matrix. 

The following peptides were studied by using standard procedures including Congo red binding analysis, TEM, inhibition assays, and cell based assays. Further, the authors also designed 60 functional fibrils using the core sequence of hA5G18.

FVFYV – amyloid fibril formation

FVFYVGGRGD – cell attachment

FVFYVGGRGD - cell attachment activity

The manuscript has merits to warrant publication. I feel, this is an important addition to the literature. However, I have couple of minor concerns before acceptance.

1.       The authors can improve the introduction section by citing some classical work of Prof. Christopher Dobson and Prof. Ulrich Hansmann.

2.       All functional peptide fibrils synthesized by the authors may be included in the supplementary material. Providing the peptides without activity may be helpful to readers for future study because authors present the peptides with activity in the current version of the publication.

Reviewer 2 Report

This is a little complicated paper, dealing with amyloid-like fibril formation and cell adhesion of short peptides that were derived from G domain of laminin. The authors synthesized various peptides with a variable amino acid sequence and investigated for their abilities with applications of various assay methods, such as Congo red staining, TEM image analysis, and HDF cell adhesion on peptide-coated plates and Sepharose beads. Each assay was carried out in a careful manner, and the results should be solid. However, the interpretation of the results seemed to be unclear and would make the readers confusing. Nevertheless, the paper includes many important findings, hence it would be worth being published. I suggest the authors to reconsider the following points and revise the manuscript.

1. In the paper, three types of base materials (chitosan matrix, plastic plate, and Sepharose bead) were referred to, and the cell adhesion behaviors were different depending on the materials. The authors should explain the reasons for this observation.

2. It is difficult to understand the difference between cell attachment and cell spreading. It seems that the difference can be seen only from the morphologies of the cells, but this is not clear.

3. Line 213, “These results suggest that the cell attachment activity of hA5G18B requires the amyloid-like fibril formation.” This is conflicting with the observation that hA5G18B did not show cell attachment (Fig. 2D) but it forms amyloid-like fibril (Fig. 1).

4. Line 249, “AG73 did not.” According to Fig. 4, it seems that AG73 promoted cell spreading.

5. In Fig. 5C, the data for FVFYVGGH and RGGFVFYV must be shown.

6. Line 309, show the definition of EF1.

7. Line 314, “These results suggest that cell attachment of FVFYVGGR, FVFYVGGK, FVFYVGGRGD, FVFYVGGRGE, and poly-R is mediated by integrins and HSPGs.” I agree that integrins mediated the cell attachment according to Fig.7 and Fig. 8, but there is no evidence for the case of HSPGs. Is it possible infer this based only on the data shown in Fig. 6?

8. Line 365, These results suggest that the amyloid-like fibril formation is required for the cell attachment activity of hA5G18.” I wonder that this is not always correct because hA5G26 does not form a fibril but is positive in the peptide-coated plate assay. So, the statement should be only for the particular case of hA5G18.

Round 2

Reviewer 2 Report

According to the answers of the authors, my concerns are all clarified. The paper is now acceptable for publication.